# Proteomic Analysis of the Major Alkali-Soluble Inca Peanut (*Plukenetia volubilis*) Proteins

**DOI:** 10.3390/foods13203275

**Published:** 2024-10-16

**Authors:** Erwin Torres-Sánchez, Esperanza Morato, Blanca Hernández-Ledesma, Luis-Felipe Gutiérrez

**Affiliations:** 1Facultad de Ciencias Agrarias, Universidad Nacional de Colombia Sede Bogotá, Carrera 30 No. 45-03, Bogotá 111321, Colombia; egtorressa@unal.edu.co; 2Centro de Biología Molecular Severo Ochoa (CBM, CSIC-UAM), Nicolás Cabrera 1, 28049 Madrid, Spain; emorato@cbm.csic.es; 3Instituto de Investigación en Ciencias de la Alimentación (CIAL, CSIC-UAM, CEI-UAM+CSIC), Nicolás Cabrera 9, 28049 Madrid, Spain; 4Instituto de Ciencia y Tecnología de Alimentos (ICTA), Universidad Nacional de Colombia Sede Bogotá, Carrera 30 No. 45-03, Edificio 500A, Bogotá 111321, Colombia

**Keywords:** SDS-PAGE, proteomic, Blast2GO, PEAKS, bioinformatics

## Abstract

Sacha Inchi (*Plukenetia volubilis*) oil press-cake (SIPC) represents a new source of proteins of high biological value, with promissory food applications. However, knowledge of these proteins remains limited. In this study, a Sacha Inchi protein concentrate (SPC) was extracted from the SIPC, and proteomic analysis was performed to identify the major alkaline-soluble proteins. The electrophoretic profile highlighted the efficacy of alkaline pH and moderate temperature to extract the major proteins, from which a group of proteins, not previously reported, were registered. LC-MS/MS analyses produced abundant high-quality fragmentation spectra. Utilizing the Euphorbiaceae database (DB), 226 proteins were identified, with numerous well-assigned spectra remaining unidentified. PEAKS Studio v11.5 software generated 1819 high-quality de novo peptides. Data are available via ProteomeXchange with identifier PXD052665. Gene ontology (GO) classification allowed the identification of sequenced proteins associated with biological processes, molecular functions, and cellular components in the seed. Consequently, the principal alkali-soluble proteins from SPC were characterized through derived functional analysis, covering 24 seed-storage-, 27 defense-, and 12 carbohydrate- and lipid-metabolism-related proteins, crucial for human nutrition due to their sulfur-containing amino acids, antioxidant properties, and oil yields, respectively. This research makes a significant contribution to the current understanding of the Sacha Inchi proteome and offers valuable insights for its potential applications in the food industry.

## 1. Introduction

One of the Sustainable Development Goals (SDGs) for nations is to reduce waste and mitigate food loss by the year 2030. The contemporary attention directed toward food waste has stimulated the development of methodologies aimed at its reduction, reuse, and recycling [1]. Meanwhile, non-communicable diseases (NCDs), including cardiovascular and chronic respiratory diseases, cancer, and diabetes, have shown significant increases in our society, becoming the leading causes of death and occupational disability. One strategy to counteract this situation involves incorporating food bioactive compounds into the diet [2]. Proteins and peptides from sources like nuts, cereals, and legumes have shown promising outcomes in reducing the incidence and mortality of NCDs such as cancer and cardiovascular diseases [3,4]. However, plant-based proteins and peptides derived from non-traditional products and by-products such as oilseed cakes have gained popularity due to the growing interest in developing eco-efficient food processes [5,6]. 

Sacha Inchi (SI) (*Plukenetia volubilis*) seed, also commonly known as ‘Inca Peanut’, belonging to the family Euphorbiaceae, is an oil-rich nut endemic to the Amazonian region of South America. Due to its significant agroindustry potential, SI has had a great expansion into Southeast Asia, South Africa, and Southwest China [7]. SI kernels are utilized for oil extraction due to the abundance of essential fatty acids ω-3 and ω-6 [8]. The primary by-product of the oil extraction process is the press-cake (SIPC) that can constitute up to 50% of the weight of the seeds [9]. SIPC contains important amounts of bioactive compounds, including proteins (40–46%), essential amino acids, minerals, and vitamins [10]. It has been used as a functional ingredient in various food products [11], as an ingredient for animal feed formulations [12,13], and as a non-conventional source of proteins, hydrolysates, and protein isolates [14,15]. 

By using the Osborne procedure, over 90% of the proteins of the SIPC may be solubilized. These proteins are sequentially classified based on the type of solvent used for their extraction, such as albumins, globulins, prolamins, and glutelins, with molecular weights (MWs) ranging from 6 to 70 kDa [16]. Our recent research demonstrates an efficient protein extraction procedure from SIPC using alkaline water (pH 11.0) and moderate temperature (~65 °C), yielding a protein concentrate (SPC) with 84.0% protein content and 24.7% extraction yield. Attenuated Total Reflectance—Fourier Transform Infrared Spectroscopy analysis revealed predominant β-sheet and β-turn structures. Also, alkaline pH facilitated sulfhydryl group conversion to disulfide bonds, imparting desirable techno-functional properties to this SPC, such as oil and water absorption capacities (6.6 ± 1.4 and 4.6 ± 0.6 *w*/*w*, respectively). However, the major proteins contained in SPC were not well characterized [17]. Therefore, the aim of this study was to characterize the principal alkali-soluble SI proteins using proteomic analysis techniques. Gel electrophoresis, LC–MS/MS, and gene ontology (GO) classification were used to obtain a better description of the SPC proteins and to classify them according to their localization and functional properties as a basis of their potential value in the food industry. 

## 2. Materials and Methods

SIPC was supplied by SumaSach’a (Mosquera, Cundinamarca, Colombia). All chemicals (reagents and solvents) were of analytical grade and provided by Sigma-Aldrich (St. Louis, MO, USA) and Merck (Kenilworth, NJ, USA). The electrophoresis analysis was conducted using equipment and reagents provided by Bio-Rad (Hercules, CA, USA).

### 2.1. Sacha Inchi Protein Concentrate (SPC) Production

SIPC adequacy and SPC production were made following the reported methodology [17]. Briefly, defatted SIPC was mixed in deionized water (1:10 ratio, *w*/*v*) under alkaline conditions (pH 11.0) with continuous stirring (1 h; 800 rpm) at 60–70 °C. The resulting supernatant was retrieved, neutralized, concentrated, and diafiltrated. Subsequently, the solution was freeze-dried and stored at −18 °C until assays. 

### 2.2. Sodium Dodecyl Sulfate–Polyacrylamide Gel Electrophoresis (SDS-PAGE)

SDS-PAGE analysis was carried out to assess the protein profile of SPC, following the methodology reported previously [18]. A Criterion^TM^ Cell system (Bio-Rad) was used. The sample was dissolved in sample buffer containing 50 mM Tris-HCl (pH 6.8), 1.6% (*w*/*v*) SDS, 0.002% (*w*/*v*) bromophenol blue, 2% (*v*/*v*) β-mercaptoethanol, and 8% (*v*/*v*) glycerol. It was then incubated at 100 °C for 5 min and centrifuged at 1100 rpm for 10 s. A 20 µL aliquot of the sample, corresponding to 75 µg of protein quantified using the Pierce Bicinchoninic Acid (BCA) kit (Thermo Fisher Scientific, Waltham, MA, USA), was loaded onto a 12% Bis-Tris Criterion™ XT Precast Gel polyacrylamide gel. The electrophoretic migration was carried out, firstly at 100 V for 5 min and then at 150 V for 1 h. The gel image was taken using the Molecular Imager^®^ Versadoc™ MP 4000 (Bio-Rad), and the image was analyzed using Image Lab 6.1 software (Bio-Rad).

### 2.3. Proteomic Analysis and Functional Annotations

#### 2.3.1. Protein Identification

Protein identification and characterization by liquid chromatography coupled to tandem mass spectrometry (LC-MS/MS) was carried out in the ‘Center of Molecular Biology Severo Ochoa (CBM) Proteomics Facility’ (CSIC, Madrid, Spain) that belongs to ProteoRed. SPC was suspended in a volume up to 50 µL of sample buffer (corresponding to 25 µg of protein) and loaded onto 1.2 cm wide wells of a conventional SDS-PAGE hand-cast gel (0.75 mm thick, 4% stacking, and 10% resolving). The run was stopped as soon as the front entered 3 mm into the resolving gel, so that the whole proteome became concentrated in the stacking/resolving gel interface. The unseparated protein bands were visualized by Coomassie staining, excised, cut into cubes (2 × 2 mm), and placed in 0.5 mL microcentrifuge tubes [19]. The gel pieces were destained in acetonitrile (ACN)/water (1:1), and disulfide bonds from cysteinyl residues were reduced with 10 mM dithiothreitol for 1 h at 56 °C, and then thiol groups were alkylated with 10 mM iodoacetamide for 30 min at room temperature in the darkness and in situ digested with sequencing grade trypsin (Promega, Madison, WI, USA) as described in the literature [20]. The gel pieces were shrunk by removing all liquid using ACN, which was pipetted out, and the gel pieces were dried in a Speedvac. The dried gel pieces were re-swollen in 100 mM Tris-HCl pH 8.0 and 10 mM CaCl_2_ with 60 ng/µL trypsin at 5:1 protein/enzyme (*w*/*w*) ratio. The tubes were kept in ice for 2 h and incubated at 37 °C for 12 h. Digestion was stopped by the addition of 1% trifluoroacetic acid (TFA). Whole supernatants were dried down. Then, they were resuspended in 50 µL of 0.1% TFA. For desalting, OMIX Pipette Tips C18 (Agilent Technologies, Santa Clara, CA, USA) were utilized, which were washed with 10 volumes of 100% ACN, followed by 10 volumes of 50% ACN. The tips were equilibrated in 0.1% TFA, and the sample was passed through the tips 10 times to ensure efficient binding. Subsequently, the tips were washed with 10 volumes of 0.1% TFA and eluted in 50 µL of a solution containing 0.1% TFA and 50% ACN, again passing the solution through the tips 10 times. After this process, the samples were dried once more until MS analysis. 

The dried, desalted protein digest was resuspended in 10 µL of 3% ACN and 0.1% formic acid (FA) (2 µg of digested peptides) and analyzed by reverse-phase (RP)-LC-MS/MS in an Easy-nLC 1200 system coupled to an ion trap LTQ-Orbitrap-Velos-Pro hybrid mass spectrometer (Thermo Scientific). The peptides were concentrated (on-line) using a 0.1 mm × 20 mm C18 RP precolumn (Thermo Scientific) and then separated using a 0.075 mm × 250 mm bioZen 2.6 µm Peptide XB-C18 RP column (Phenomenex, Torrance, CA, USA) operating at 0.25 μL/min. Peptides were eluted using a 180 min dual gradient. The gradient profile was set as follows: 5–25% solvent B for 135 min, 25−40% solvent B for 45 min, 40−100% solvent B for 2 min, and 100% solvent B for 18 min (solvent A: 0.1% FA in water and solvent B: 0.1% FA, 80% ACN in water). ESI ionization was performed using a Nano-bore emitters Stainless Steel ID 30 μm (Proxeon, Odense, Denmark) interface at 2.1 kV spray voltage with S-Lens of 60%. The Orbitrap resolution was set at 30,000 [21]. Peptides were detected in survey scans from 400 to 1600 amu (1 μscan), followed by twenty data-dependent MS/MS scans (Top 20), using an isolation width of 2 u (in mass-to-charge ratio units), a normalized collision energy of 35%, and dynamic exclusion applied for 60 s periods. Charge-state screening was enabled to reject unassigned and singly charged protonated ions. 

For protein and peptide identification, two complementary methodologies were employed: for proteins (1), match to a “Eurphorbiaceae” database (DB) and for peptides (2), de novo sequencing, utilizing the robust software PEAKS Studio v11.5 search tool (Bioinformatics Solutions Inc., Waterloo, ON, Canada). DB search was performed against Uniprot-eurphorbiaceae.fasta (155614 entries; UniProt free release on 11/2023; https://shorturl.at/XJFWn (accessed on 28 November 2023)) (decoy-fusion DB). The following constraints were used for the searches: tryptic cleavage after arginine and lysine (semi-specific), up to two missed cleavage sites, and tolerances of 20 ppm for precursor ions and 0.6 Da for MS/MS fragment ions, and searches were performed allowing optional methionine oxidation and cysteine carbamidomethylation. False discovery rates (FDRs) for peptide spectrum matches (PSMs) and for protein were limited to 1%. Only those proteins with at least two unique peptides being discovered from LC/MS/MS analyses were considered reliably identified [22,23]. For the de novo sequencing strategy, candidates with % confidence > 85 were accepted. 

The mass spectrometry proteomics data were deposited into the ProteomeXchange Consortium via the PRIDE [24] partner repository with the dataset identifier PXD052665 and 10.6019/PXD052665.

#### 2.3.2. Gene Ontology (GO) Classification

After the SPC protein sequences were identified using PEAKS, we used the Functional Analysis Module of OmicsBox v3.1.11 (BioBam Bioinformatics S.L., Valencia, Spain) search engine to run a functional annotation pipeline using Blast2GO methodology [25]. Then, the sequences identified were aligned employing the Basic Local Alignment Search Tool (BLAST) in Blastp-fast mode with default parameters in a cloud-based environment. Using OmicsBox, the BLAST search results were employed for subsequent analyses, including ontology annotations, filtered by taxonomy up to the Viridiplantae kingdom. After CloudBlast, the GO terms were categorized into biological process (BP), molecular function (MF), and cellular component (CC).

## 3. Results and Discussion

### 3.1. Electrophoretic Profile

SDS-PAGE was chosen to characterize SI proteins due to its effectiveness in assessing both the protein yield and the quality of polypeptide separation from proteins extracted from the SPC. Additionally, this method is compatible with downstream gel-based protein analyses, including mass spectrometry [26]. The protein profile of SPC was visualized by SDS-PAGE (Figure 1). 

Bands corresponding to fifteen proteins in the 4.2–186.9 kDa range were observed. The utilization of water (pH 11.0) and moderate temperature (60–70 °C) demonstrated to be highly efficient for the extraction of SI proteins compared to the Osborne method [25,27]. It allowed extracting proteins such as albumins (◊) (including the notable 3S albumin ‘Inca Peanut Albumin’, a dimeric, glycosylated, and basic protein with a MW of 27.5 kDa); globulins (*) (47.4, 37.7, and 21.0 kDa); prolamins (▪) (10.0, 6.3, and 4.2 kDa); and glutelins (+) (69.9 kDa), and all of them are reported in the literature [16,28]. Additionally, a group of high-MW polypeptides (¬) (between 186.9 and 86.8 kDa), which had not been previously reported, were found. These proteins could correspond to a group of glutelins with enzymatic functions such as beta-galactosidase activity, responsible for breaking down reserve polysaccharides during plant growth, and neutral ceramidase, which plays a crucial role in lipid metabolism by hydrolyzing neutral ceramides into sphingosine and free fatty acids (Table 1). That hypothesis should be corroborated with further studies. In other pulse seeds, glutelin proteins with high-MWs such as vicilins have been identified in pea (155 kDa), red bean (173 kDa), cowpea (133–140 kDa), mung bean (162 kDa), kidney bean (136–150 kDa), and faba beans (163 kDa) [29].

### 3.2. Proteomic Analysis, Gene Ontology (GO) Annotations, and Functional Analysis

#### 3.2.1. Proteomic Analysis

Proteins from SPC were profiled using LC-MS/MS, employing a mono-enzymatic digestion approach for data acquisition (classical bottom-up). From raw data using PEAKS Studio v11.5 SPIDER module search tool [30], a total of 950 PSMs, 946 scans (tandem mass scans that have a peptide spectrum associated with them), 437 features, 401 peptide sequences with modifications not including isoleucine/leucine differentiation, and 263 peptide sequences without modifications and isoleucine/leucine differentiation, were identified. Sequences constituted the key for homology-based protein identification, and PEAKS proposed two classifications: protein groups and top proteins. Protein groups represented the number of groups of proteins with significant peptides that met specified filtering criteria. Top proteins were those corroborated by the largest count of unique peptides within the protein groups [31]. A total of 226 unique sequences excluding isoleucine/leucine and post-translational modification (PTM) duplications, in 36 groups with significant peptides (≥2), were identified (Appendix A). Of these 226 unique sequences, 201 sequences represented a top protein group (Figure 2a), offering novel insights into the protein composition of *P. volubilis* kernels.

A total of 52% of the total identified proteins were homologous to proteins of *Jatropha curcas* (18.1%), *Ricinus communis* (11.9%), *Manihot esculenta* (11.1%), *Hevea brasiliensis* (9.3%), and *Vernicia fordii* (1.8%). Three protein sequences belonged to *P. volubilis* (1.3%), which have been reported in previous studies [28,32]. The remaining 46.5% of the sequences had been identified in other species of the Euphorbiaceae family. Each of these species had ≤ 2 protein sequences (Figure 2b). In general, the limited count of proteins and peptides identified by the DB search mode should be attributed to the incompleteness of SI proteome sequencing. 

Additionally, PEAKS conducted de novo sequencing directly from MS/MS data. It calculated the optimal sequence from all feasible amino acid combinations and computed peptides whose ions correspond to as many high-abundance peaks in the spectrum as possible [33]. In this study, 1819 peptides with a high confidence value (>85%) were generated (Appendix A), ranging from 6 to 18 amino acids and ~671 to ~1980 Daltons (Figure 2c,d), which represented de novo sequence tags that were not present in the DB. To the best of our knowledge, this is the first report of alkali-soluble proteins fragmented into peptides and oligopeptides, resulting in the generation of a confident peptidome. It contributes to studies aimed at sequencing proteins in this oilseed. Additionally, it adds to the studies on SI compounds, which have recently begun to develop, generating information on bioactive peptides [34,35,36] with potential applications in the food and pharmaceutical industries. 

#### 3.2.2. Gene Ontology (GO) Annotations in Proteins

The distribution of the sequenced proteins of SPC associated with the top terms of functional classification of GO categories under BP, CC, and MF are summarized in Figure 3. 

The complete list is found in Appendix A. A total of 224 (98.24%) protein sequences were assigned to 127 GO annotations: 25 to the CC, 57 to the MF, and 45 to the BP categories. In the CC category, chloroplast (GO:0009507), cytoplasm (GO:0005737), membrane (GO:0016020), and nucleus (GO:0005634) were the predominant terms, with 88, 45, 20, and 17 protein sequences assigned, respectively. In the MF category, magnesium ion binding (GO:0000287), monooxygenase activity (GO:0004497), and ribulose bisphosphate carboxylase activity (GO:0016984) were the main distributed terms with 85 protein sequences assigned to each category, followed by ATP binding (GO:0005524) and ATP-dependent protein-folding chaperone (GO:0140662), with 55 and 37 protein sequences assigned, respectively, and translation elongation factor activity (GO:0003746), GTPase activity (GO:0003924) and GTP binding (GO:0005525) with 36 protein sequences assigned to each one. In the BP category, photorespiration (GO:0009853) and reductive pentose phosphate cycle (GO:0019253) were the predominant terms with 85 protein sequences assigned to each one, followed by translational elongation (GO:0006414) and protein refolding (GO:0042026) with 36 protein sequences and chaperone cofactor-dependent protein refolding (GO:0051085), response to stress (GO:0006950), RNA processing (GO:0006396), and protein localization to plasma membrane (GO:0072659), with 30, 26, 20, and 20 protein sequences assigned, respectively. 

#### 3.2.3. Protein Functional Analysis

GO analysis was performed to elucidate the roles of major alkali-soluble proteins and their biological significance in the SI seed proteome [37]. In this research, our aim was to highlight sequenced proteins identified in the SPC that could potentially be useful for the food industry through functional analysis. In *P. volubilis*, the nutrients deposited in the seeds are essential for the plant development. We identified 24 seed proteins associated with the ‘nutrient reservoir activity’ GO term (GO:0045735) (Table 2 and Appendix A). 

This GO term included three protein species (PS) of Legumin B and Legumin A, 14 Cupin type-1 domain-containing proteins, two Glutelin type-A 3 PS, and one PS each of nutrient reservoir and 11S globulin subunit beta. The relationship among the storage protein Legumin B, the precursor Legumin A, and the 11S globulin subunit beta, with MWs between 43 and 53 kDa, is based on the composition of each SI legumin PS by two subunits. These subunits consist of an acidic (α-subunit, 30–40 kDa) and a basic (β-subunit, 20 kDa) unit, covalently joined through a single disulfide bond. These chains are assembled within the protein bodies, yielding mature forms that are deposited in a particular temporal order within 7S (trimeric) and 11-12S globulins (hexameric), as previously observed in another oilseed species like *J. curcas* [38]. SDS-PAGE analysis provided strong evidence supporting this hypothesis for major *P. volubilis* proteins, because experiments (1D and 2D) revealed that despite similar mobility under native conditions, the PS dissociate into individual subunits under reducing conditions, indicating the presence of α-β subunits linked by disulfide bridges [16,39,40]. 

Glutelin type-A 3 and Cupin type-1 domain-containing PS are also seed-storage-related proteins. Glutelin type-A 3 belongs to the glutelin family, commonly found in plants like rice and other cereals. It exists as a ~54 kDa proglutelin, composed of two polypeptides, a ~35 kDa acidic (α-polypeptide), and a ~22 kDa basic (β-polypeptide), linked by a disulfide bond. Interestingly, glutelin type-A 3 is encoded by a multigene family, as seen in rice with the type-A3 gene itself [41], where its abundance is a food quality determinant among ecotypes. These proteins play essential roles in various aspects of seed development, potentially including the formation of specific tissues like the endosperm and embryo [38]. Likewise, the fourteen Cupin type-1 domain-containing proteins, with MWs between 25 and 83 kDa, identified in this study belonged to the storage protein/sucrose protein-binding group, a type of globulin implicated in the sucrose uptake system of legumes [42]. Notably, cupins of leguminous and non-leguminous plants play a multifaceted role. Firstly, they contribute to the dietary protein value by being a rich source of sulfur-containing amino acids. Secondly, these cupins serve as the primary nitrogen source for the developing plant itself. Additionally, they play a role in the plant’s defensive function against insect predation [38]. As a consequence of these mentioned roles, the observed higher levels of sulfur-containing amino acids, specifically methionine + cysteine, tyrosine, threonine, and tryptophan in SI kernels compared to other oilseeds like soybean, peanut, cottonseed, and sunflower [43] could be attributed to the abundance and role of cupin PS. This makes SPC particularly valuable for foods due to their ability to fulfill human nutritional needs. 

A group of defense-related protein sequences, like Heat shock protein 70 and Heat shock PS, Ubiquitin C variant sequence, Ubiquitin, Ubiquitin-like domain-containing protein, Elongation factor 1-alpha, and Cystatin domain-containing protein, were also identified (Table 3). 

The sequenced heat shock protein family, with MWs between 73 and 75 kDa, as well as the five uncharacterized proteins homologous to *Hevea brasiliensis* and *Manihot esculenta* species (Appendix A), were associated with the ‘heat shock protein binding’, ‘2-alkenal reductase [NAD(P)+] activity’, and ‘ATP-dependent protein—folding chaperone’ GO terms (GO:0031072, GO:0032440, and GO:0140662, respectively). Structurally, Heat shock protein 70 (Hsp70) consists of a highly conserved N-terminal ATPase domain of 44 kDa and a C-terminal peptide-binding domain of ~25 kDa. They play a role in protecting cells against stress and promoting cell survival under adverse conditions. An analysis of the *Arabidopsis* and spinach Hsp70 genes demonstrated that Hsp70 chaperones are expressed in response to stress conditions such as heat, cold, drought, and other stresses [44]. Additionally, the Hsp70 extracted from *Croton cajucara* B. has been demonstrated to have a gastro-protective effect against ethanol-induced gastric ulcers in male Wistar rats [45]. 

The Ubiquitin-like domain-containing protein, Ubiquitin C variant, and Ubiquitin (fragment) PS, with MWs between 14 and 34 kDa, were associated with multiple GO terms (GO:0031386, GO:0003729, GO:0003677, GO:0046872, or GO:0031625) in the MF category like the ‘protein tag activity’, ‘mRNA binding’, ‘DNA binding’, ‘metal ion binding’, or ‘ubiquitin protein ligase binding’, respectively. The low-MW PS, around 14 or 17 kDa, represent free mono-ubiquitins whose function was protein tagging activity. Structurally, higher-MW ubiquitins can be conjugated with other proteins for their catabolic process, and they play a fundamental role in proper protein folding, protein turnover, DNA repair mechanisms, and progression through the cell cycle [46]. Ubiquitin proteins protect the embryos of mature *J. curcas* seeds against environmental stress by degrading potentially harmful proteins, thereby regulating the concentration of hormones and regulatory proteins to maintain cellular homeostasis [38]. 

Thirty-three PS of Elongation Factor 1-alpha (EF-1α) were identified, with MWs ranging between 17 and 49 kDa (only the top 5 are shown in Table 3, and the remaining PS are found in Appendix A). It is important to note that EF-1α plays a crucial role in translational elongation by facilitating RNA binding. It has been associated with diseased tissues, including rust-infected leaves [47]. The abundance of these proteins in the SPC may potentially be attributed to the challenging environmental conditions of the Amazon rainforest, where the plant has evolved. In this regard, Cystatin domain-containing proteins, with a MW of ~13 kDa, are potent cysteine protease inhibitors essential for maintaining proteostasis in plants. They bind to and block the active site of cysteine proteases, thereby preventing their hydrolytic cleavage [48]. Also, they play a crucial role in plant defense by inhibiting cysteine proteases produced endogenously in response to biotic and abiotic stresses. Additionally, they may offer protection against external threats like phytophagous insects, root nematodes, and fungal pathogens [49]. 

SI has emerged as a novel oilseed crop due to its elevated oil content and distinctive profile of essential fatty acids, leading to industrialization. Carbohydrate- and lipid-metabolism-related proteins were also identified (Table 4). The cytosolic-abundant Glyceraldehyde-3-phosphate dehydrogenase (GAPDH) protein sequences with a unique peptide found in this study (data found in the Peaks Project at the ProteomeXchange repository) were associated with their specific enzymatic activity GO term (GO:0004365), which catalyze the formation of 1,3-bisphosphoglycerate from Gly-3-P (G3P) in *J. curcas* and others oilseed species. The specific expression of GAPDH in the species *J. curcas*, which belongs to the Euphorbiaceae family, resulted in a 3–4-fold increase in G3P, which ensured improved oil deposition [38]. This suggested that it might constitute the essential protein group responsible for significant oil deposition in SI, accounting for up to 60% of the kernel’s weight [8]. 

The synthesis of lipids requires carbon, energy, and reducing equivalents, which are supplied directly or indirectly via glycolysis. We found 85 PS of the ribulose bisphosphate carboxylase large chain, with MWs ranging between ~46 and ~53 kDa (only the top 5 are displayed in Table 4, while the remaining PS can be found in Appendix A). Ribulose-1,5-bisphosphate carboxylase/oxygenase (RuBisCO) is the most abundant soluble protein found among the three domains of life—archaea, bacteria, and eukarya. The ribulose bisphosphate carboxylase large chain is one of the two subunits that compose this enzyme. RuBisCO catalyzes ribulose-1,5-bisphosphate carboxylation in the Calvin cycle, facilitating CO_2_ fixation and carbohydrate biosynthesis in photoautotrophs [50]. Fatty acids combine with glycerol to form triacylglycerols, which gather into oil bodies. These contain triacylglycerols at the core, surrounded by a phospholipid monolayer with proteins like oleosins, regulating oil body size and lipid buildup in seeds [51]. We identified Oleosin 1, Oleosin 2, and Oleosin 3 that belong to *P. volubilis* with MWs ranging from 14 to 26 kDa (Table 4). Oleosins are small alkaline proteins of 15–26 kDa, which represent 0.5–4% (*w*/*w*) of the oil body. Oleosins have a central hydrophobic domain of about 70 non-polar residues, surrounded by polar N-terminal and C-terminal amphipathic domains [51]. This could explain why the SPC obtained through alkaline extraction exhibits excellent oil absorption capacity and water absorption index properties, while being deficient in water solubility index values and foam properties, as previously shown [17].

## 4. Conclusions

This study makes a significant contribution by identifying and characterizing unknown alkaline-soluble proteins using LC-MS/MS techniques and bioinformatics tools. The results reveal a unique diversity of proteins related to nutrient storage, plant defense response, and biosynthesis of lipid and carbohydrate functions. To the best of our knowledge, this is a pioneer report of de novo peptide and protein sequences of SI with a high confidence level, providing relevant information for consolidating the proteome of this oilseed. Also, the findings constitute a conceptual basis for future studies related to the release of bioactive peptides through in vitro and/or in vivo hydrolysis with potential applications in the food industry. 

## Figures and Tables

**Figure 1 foods-13-03275-f001:**
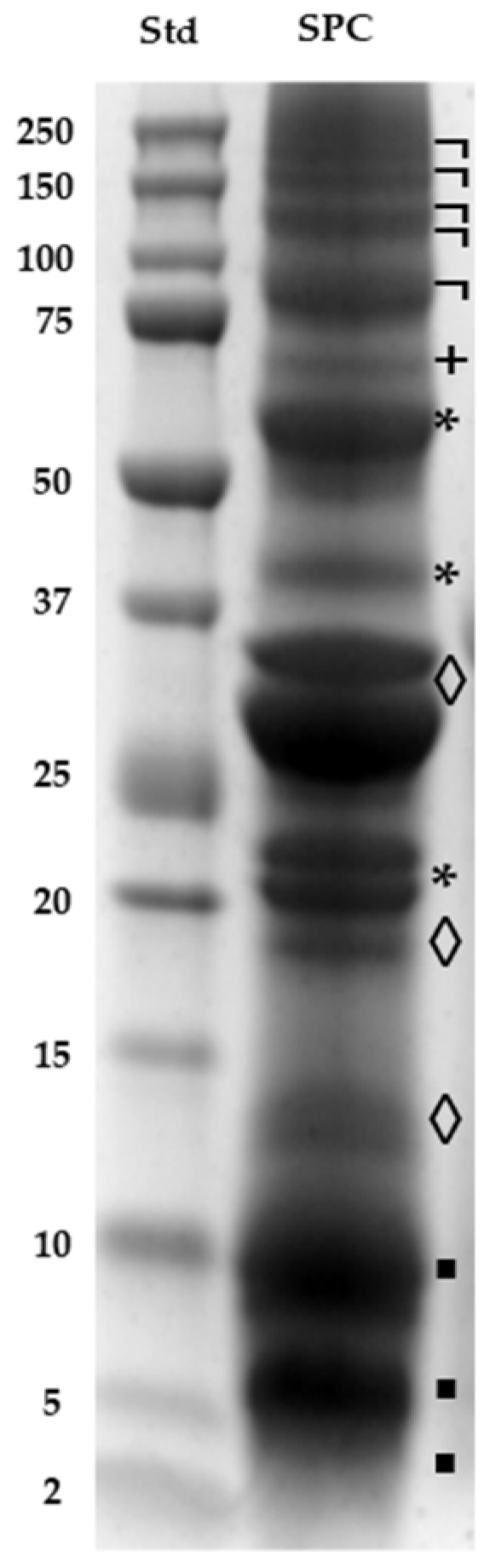
Electrophoretic (SDS-PAGE) analysis. (Std) Line with Precision Plus Protein^TM^ Dual Xtra Standard. Sacha Inchi protein concentrate (SPC). Note the different polypeptides in lines classified into albumins (◊), globulins (*), prolamins (▪), glutelins (+), and higher-molecular-weight polypeptides (¬).

**Figure 2 foods-13-03275-f002:**
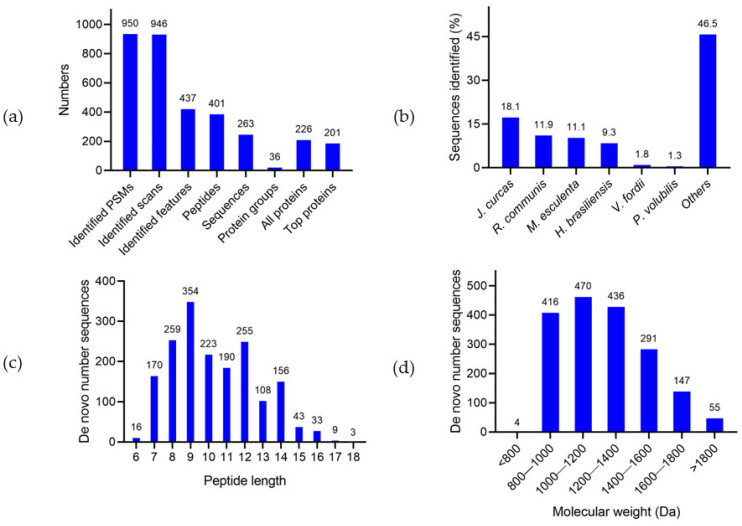
(**a**) Protein identification statistics; (**b**) principals identified species matches in database (DB); (**c**) length distribution of de novo peptides; (**d**) molecular weight (Da) of de novo peptides.

**Figure 3 foods-13-03275-f003:**
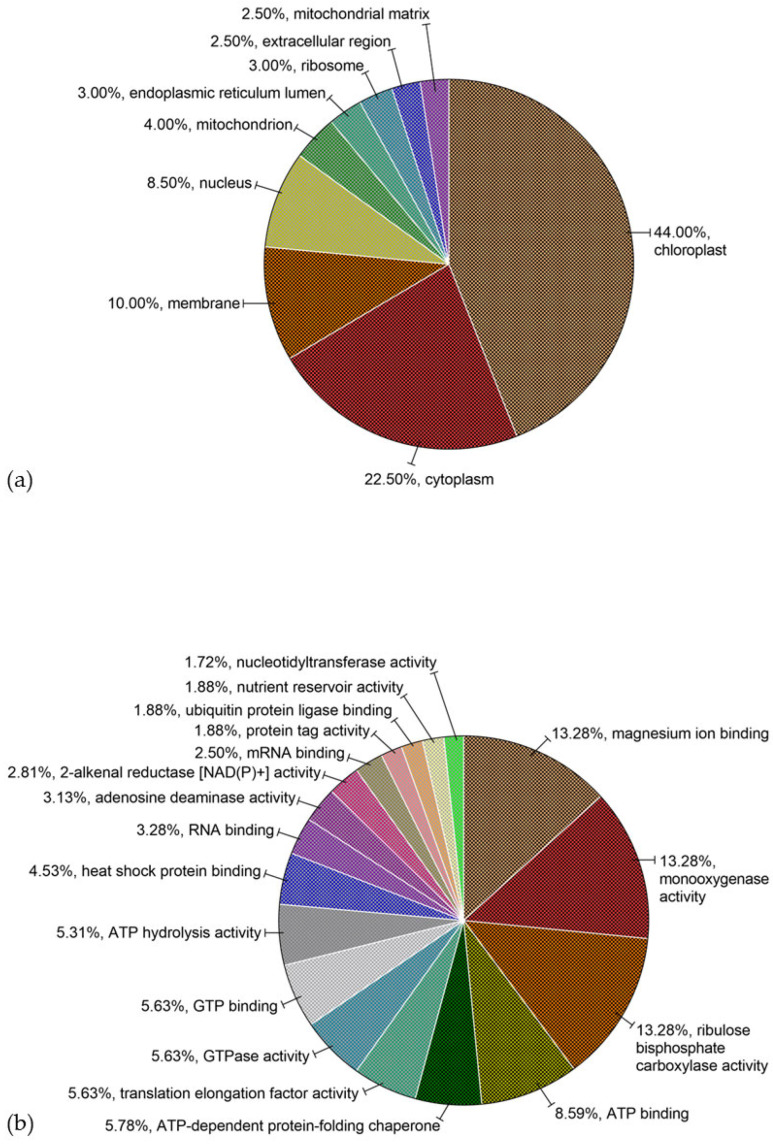
Percentage of identified proteins associated with terms of functional classification of gene ontology (GO) categories under (**a**) cellular component, CC; (**b**) molecular function, MF; and (**c**) biological process, BP.

**Table 1 foods-13-03275-t001:** Glutelin proteins extracted from Sacha Inchi (*Plukenetia volubilis*) press-cake protein concentrate (SPC).

Accession ^a^	Specie ^b^	Unique ^c^	Average Mass ^d^	Description ^e^
A0A6A6LG23	*Hevea brasiliensis*	2	182,269.31	Beta-galactosidase
A0A2C9W8Z9	*Manihot esculenta*	2	100,566.17	Beta-galactosidase
B9S377	*Ricinus communis*	2	85,277.63	Neutral ceramidase
A0A2C9WL90	*Manihot esculenta*	2	85,249.52	Neutral ceramidase
A0A2C9WFX1	*Manihot esculenta*	2	85,241.37	Neutral ceramidase
A0A6A6MFZ7	*Hevea brasiliensis*	2	84,464.71	Neutral ceramidase
A0A6A6LBT4	*Hevea brasiliensis*	2	74,518.06	Uncharacterized protein
A0A6A6MIV4	*Hevea brasiliensis*	2	73,365.3	Neutral ceramidase
A0A2C9W9U3	*Manihot esculenta*	2	73,318.14	Uncharacterized protein

^a^ Accession number of the protein as seen in the FASTA database. ^b^ Specie homologous in database search. ^c^ Number of high-confidence supporting peptides that were mapped to only one protein group. ^d^ Protein mass calculated using the average mass (Da). ^e^ Protein’s header information as seen in the FASTA database.

**Table 2 foods-13-03275-t002:** Alkali-soluble seed storage proteins extracted from Sacha Inchi (*Plukenetia volubilis*) press-cake protein concentrate (SPC).

Accession ^a^	−10LgP ^b^	Coverage (%) ^c^	Peptides ^d^	Unique ^e^	Average Mass ^f^	Description ^g^
Q9M4Q8	306.11	35.08	20	1	53,668	Legumin B, putative
B9SF36	304.76	25.59	21	1	57,338	Legumin A, putative
B9SDX6	302.79	33.82	19	1	53,652	Legumin B, putative
A0A067K2P3	254.89	26.88	17	10	54,585	Cupin type-1 domain-containing protein
B9SF35	250.92	15.16	13	6	53,490	Legumin A, putative
B9T5E7	248.94	20.32	12	5	55,970	Glutelin type-A 3, putative
A0A2C9UPV6	244.61	15.55	12	6	53,821	Cupin type-1 domain-containing protein
A0A2C9UTB0	219.39	12.68	9	4	53,609	Cupin type-1 domain-containing protein
A0A6A6KGZ5	205.96	14.55	8	1	49,303	Cupin type-1 domain-containing protein
A0A6A6KD91	205.42	14.16	8	1	49,020	Cupin type-1 domain-containing protein
B9T1B8	202.98	15.62	7	2	51,163	Legumin A, putative
R4I518	192.83	18.31	7	4	46,572	Legumin B (Fragment)
A0A067K3Z1	172.17	8.61	5	1	52,443	Cupin type-1 domain-containing protein
R4I3K1	171.53	8.03	5	2	55,864	Glutelin type-A 3 (Fragment)
A0A2C9UDP9	166.13	6.76	4	1	56,598	Cupin type-1 domain-containing protein
A0A067KW23	162.61	7.09	5	2	57,166	Cupin type-1 domain-containing protein
A0A2C9W805	144.81	6.82	3	1	57,672	Cupin type-1 domain-containing protein
B9S9Q7	140.58	11.40	4	2	43,597	11S globulin subunit beta, putative
B9SKF4	131.46	6.94	4	1	40,099	Nutrient reservoir, putative
A0A2C9UDV1	102.04	4.58	2	1	29,405	Cupin type-1 domain-containing protein
A0A6A6KD87	65.72	6.82	1	1	25,130	Cupin type-1 domain-containing protein
A0A067KVT0	65.29	1.75	1	1	66,360	Cupin type-1 domain-containing protein
A0A6A6NJ14	63.36	1.52	1	1	82,874	Cupin type-1 domain-containing protein
A0A6A6NIY9	63.36	2.24	1	1	55,956	Cupin type-1 domain-containing protein

^a^ Accession number of the protein as seen in the FASTA database. ^b^ PEAKS protein score (−10lgP) calculated as the weighted sum of the −10lgP scores of the protein’s supporting peptides. ^c^ Percentage of the protein sequence that was covered by supporting peptides. ^d^ Number of high-confidence supporting peptides. ^e^ Number of high-confidence supporting peptides that were mapped to only one protein group. ^f^ Protein mass calculated using the average mass (Da). ^g^ Protein’s header information as seen in the FASTA database. Note: The fragmentation spectra of protein sequences with only a single unique peptide were manually reviewed and validated to confirm their suitability.

**Table 3 foods-13-03275-t003:** Alkali-soluble defense-related proteins extracted from Sacha Inchi (*Plukenetia volubilis*) press-cake protein concentrate.

Accession ^a^	−10LgP ^b^	Coverage (%) ^c^	Peptides ^d^	Unique ^e^	Average Mass ^f^	Description ^g^
A0A6A6LLV6	173.49	12.67	6	4	71,856	Uncharacterized protein
B9RGN2	173.49	12.69	6	4	71,610	Heat shock protein, putative
A0A2C9V090	173.49	12.67	6	4	71,884	Uncharacterized protein
B2MW33	173.49	12.67	6	4	71,738	Heat shock protein 70
A0A6A6LJD5	173.49	14.54	6	4	63,321	Uncharacterized protein
B9RGN3	173.49	12.67	6	4	71,867	Heat shock protein, putative
A0A2C9W9U3	159.70	6.63	4	2	73,318	Uncharacterized protein
A0A6A6LBT4	159.70	6.56	4	2	74,518	Uncharacterized protein
A0A067JQG5	97.24	11.69	2	2	17,247	Ubiquitin-like domain-containing protein
A0A6A6L0J0	97.24	6.06	2	2	33,548	Ubiquitin-like domain-containing protein
A0A2C9UND2	97.24	11.54	2	2	17,702	Ubiquitin-like domain-containing protein
A0A6A6KXT8	97.24	6.47	2	2	31,580	Ubiquitin-like domain-containing protein
A0A067KV44	97.24	5.84	2	2	34,484	Ubiquitin-like domain-containing protein
A0A6A6L0Z6	97.24	5.10	2	2	39,593	Ubiquitin-like domain-containing protein
A0A6A6N2G6	97.24	6.79	2	2	30,429	Ubiquitin-like domain-containing protein
D6BR61	97.24	14.06	2	2	14,673	Ubiquitin C variant
A0A6A6KMK8	97.24	14.06	2	2	14,673	Ubiquitin-like domain-containing protein
J3SA20	97.24	10.23	2	2	19,771	Ubiquitin (Fragment)
A0A6A6NCA8	97.24	3.76	2	2	53,550	Ubiquitin-like domain-containing protein
A0A6A6NCR1	97.24	3.47	2	2	58,001	Ubiquitin-like domain-containing protein
A0A6A6LEU2	96.25	4.44	2	2	47,446	Elongation factor 1-alpha
B9TK78	96.25	7.17	2	2	28,942	Elongation factor 1-alpha, putative (Fragment)
A0A6A6LG04	96.25	4.28	2	2	49,009	Elongation factor 1-alpha
A0A0H3YHL9	96.25	4.23	2	2	49,522	Elongation factor 1-alpha
Q5VBE3	96.25	11.52	2	2	17,642	Elongation factor 1-alpha
A0A067JPF4	62.90	13.51	1	1	12,240	Cystatin domain-containing protein
A0A067JQS0	62.90	13.51	1	1	12,155	Cystatin domain-containing protein

^a^ Accession number of the protein as seen in the FASTA database. ^b^ PEAKS protein score (−10lgP) calculated as the weighted sum of the −10lgP scores of the protein’s supporting peptides. ^c^ Percentage of the protein sequence covered by supporting peptides. ^d^ Number of high-confidence supporting peptides. ^e^ Number of high-confidence supporting peptides mapped to only one protein group. ^f^ Protein mass calculated using the average mass (Da). ^g^ Protein’s header information as seen in the FASTA database. Note: The fragmentation spectra of protein sequences with only a single unique peptide were manually reviewed and validated to confirm their suitability.

**Table 4 foods-13-03275-t004:** Carbohydrate- and lipid-metabolism-related proteins extracted from Sacha Inchi (*Plukenetia volubilis*) press-cake protein concentrate.

Accession ^a^	−10LgP ^b^	Coverage (%) ^c^	Peptides ^d^	Unique ^e^	Average Mass ^f^	Description ^g^
Q9FSJ2	156.38	27.80	5	2	22,915	Superoxide dismutase (fragment)
Q9STB5	156.38	27.80	5	2	22,915	Superoxide dismutase (fragment)
A0A2C9VLE6	156.38	24.46	5	2	25,865	Superoxide dismutase
A0A067K4A7	133.61	14.54	4	1	36,668	Glyceraldehyde-3-phosphate dehydrogenase
J9S4Z6	115.99	20.41	3	3	15,767	Oleosin2
A0A8A4JBY4	95.49	4.77	2	2	53,346	Ribulose bisphosphate carboxylase large chain
A7BGA7	95.49	5.19	2	2	48,923	Ribulose bisphosphate carboxylase large chain
A0A2I6R6R7	95.49	5.23	2	2	48,712	Ribulose bisphosphate carboxylase large chain
Q3T5B9	95.49	4.95	2	2	51,525	Ribulose bisphosphate carboxylase large chain
E0D956	95.49	5.45	2	2	46,624	Ribulose bisphosphate carboxylase large chain
J9RZJ1	64.51	7.91	1	1	14,862	Oleosin3
J9S7S3	63.76	6.29	1	1	15,279	Oleosin1

^a^ Accession number of the protein as seen in the FASTA database. ^b^ PEAKS protein score (−10lgP) calculated as the weighted sum of the −10lgP scores of the protein’s supporting peptides. ^c^ Percentage of the protein sequence covered by supporting peptides. ^d^ Number of high-confidence supporting peptides. ^e^ Number of high-confidence supporting peptides mapped to only one protein group. ^f^ Protein mass calculated using the average mass (Da). ^g^ Protein’s header information as seen in the FASTA database. Note: The fragmentation spectra of protein sequences with only a single unique peptide were manually reviewed and validated to confirm their suitability.

## Data Availability

The data presented in this study are available on request from the corresponding author due to restrictions of the project.

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
