# Peer review of "Proteomic Analysis of the Major Alkali-Soluble Inca Peanut (*Plukenetia volubilis*) Proteins"

_foods, 2024, doi:10.3390/foods13203275_

Round 1

Reviewer 1 Report

Comments and Suggestions for Authors

The manuscript “Proteomic analysis of the major alkali-soluble Sacha Inchi (Plukenetia volubilis) protein” is a well organised study in terms of utilisation of protein as human food from a neglected source.

Addressing the following comments hopefully improve the manuscript.

Please mention the common name of Sacha Inchi (Plukenetia volubilis) “Inca Peanut” in the title.

Please consider revising “belonging to the Euphorbiaceae family” with “(family: Euphorbiaceae)”.

Please consider revising “However, its major proteins were not identified.” with “However, the major proteins were not well characterised.”.

Please detail the “Eurphorbiaceae” database, including url, access date, if openly accsible etc. Similarly, please detail also the PEAKS Studio v11.5 SPIDER search tool, 

Please mention why SDS PAGE was chosen to characterise proteins, level the Gel image according to, follow/cite DOI:https://doi.org/10.3390/agronomy11010107 [not like just (1)].

Please mention why GO study was carried out.

Please add a single sentence background of the study at the beginning of the conclusion.

Author Response

Reviewer 1

The manuscript “Proteomic analysis of the major alkali-soluble Sacha Inchi (Plukenetia volubilis) protein” is a well organised study in terms of utilisation of protein as human food from a neglected source.

Addressing the following comments hopefully improve the manuscript.

  1. Please mention the common name of Sacha Inchi (Plukenetia volubilis) “Inca Peanut” in the title.

Answer: The common name “Inca peanut” has been added to the title as suggested. Additionally, we have modified the sentence in line 47 according to the suggestion of the reviewer.

  1. Please consider revising “belonging to the Euphorbiaceae family” with “(family: Euphorbiaceae)”.

Answer: As suggested, it has been revised (line 48).

  1. Please consider revising “However, its major proteins were not identified.” with “However, the major proteins were not well characterised.”.

Answer: The sentence has been modified as suggested (line 68-69).

  1. Please detail the “Eurphorbiaceae” database, including url, access date, if openly accsible etc. Similarly, please detail also the PEAKS Studio v11.5 SPIDER search tool,

Answer: The URL for the Euphorbiaceae database was added (line 148), together the word 'free' to specify that it is an open-access database. The access date was November 2023, previously indicated as '11/2023' in the same line.

In addition, the word 'module' was added (line 209).

  1. Please mention why SDS PAGE was chosen to characterise proteins, level the Gel image according to, follow/cite DOI:https://doi.org/10.3390/agronomy11010107 [not like just (1)].

Answer: As suggested, the Figure 1 was modified. Also, the description of the reasons why SDS-PAGE was used has been included (lines 172-175).

  1. Please mention why GO study was carried out. See/cite this publication https://doi.org/10.1371/journal.pone.0253384

Answer: As suggested, the explanation why GO study was carried out has been included (lines 274-275).

  1. Please add a single sentence background of the study at the beginning of the conclusion.

Answer: As suggested, the sentence has been included (lines 420-421).

Reviewer 2 Report

Comments and Suggestions for Authors

Comments for the Authors

This article describes proteomic analysis of alkaline extracts of Sacha Inchi oil-press cake. The authors show extraction efficiency with SDS-PAGE separation of proteins in the extract. The extract was digested with trypsin and analyzed by LC-MS/MS. They were able to identify a number of proteins associated with specific GO terms and several were assigned to seed storage, defense, stress response, and carbohydrate / lipid metabolism.

The authors describe a gap in knowledge applicable and relevant to readers of Foods journal. The paper is well written with a good experiment rationale and background, though the amount of data included is on the lighter side (is the entire manuscript based on a single MS injection? If not, details should be included). Some of the experimental details are missing and need to be included prior to acceptance. Finally, this reviewer noticed a couple of inappropriate citations that need to be corrected. Additional minor revisions as detailed below.

Recommendation to editor: Accept with minor revisions.

Minor Points:

1.     Line 63: I would prefer the reference to the earlier work go after this sentence.

2.     Line 66: Could the authors define “desirable techno-funcitonal properties” or list some of said properties in the manuscript?

3.     Line 86: How was the freeze-dried sample reconstituted – in what buffer? How was the protein concentration determined – with what techniques / instrumentation? These are required details.

4.     Line 96: What is the composition of the “sample buffer” indicated here?

5.     Line 102: Ref [19] is inappropriate here. There are better citations describing this technique.

6.     Line 111: Again after drying, what were the samples reconstituted in? and when desalted how was that done? In steps, with washes? What eluent? Volume? I assume they were redried again after desalting but that information is also missing.

7.      Line 113: How much of the sample was injected – representing how many ug of protein digest? How many injections total?

8.     Line 125: Ref [21] is inappropriate here as it doesn’t fit to support the selection of orbitrap resolution.

9.     Line 229: Define the acronyms BP, CC, and MF in the text here.

10.  Line 306: “to fulfill the nutritional needs human nutrition” change to “to fulfill human nutritional needs”

11.  Line 364: Sentence beginning here needs to be rewritten to make it clear the authors are describing previous results

12.  Line 396: delete “demonstrated”

Author Response

Reviewer 2

This article describes proteomic analysis of alkaline extracts of Sacha Inchi oil-press cake. The authors show extraction efficiency with SDS-PAGE separation of proteins in the extract. The extract was digested with trypsin and analyzed by LC-MS/MS. They were able to identify a number of proteins associated with specific GO terms and several were assigned to seed storage, defense, stress response, and carbohydrate / lipid metabolism.

The authors describe a gap in knowledge applicable and relevant to readers of Foods journal. The paper is well written with a good experiment rationale and background, though the amount of data included is on the lighter side (is the entire manuscript based on a single MS injection? If not, details should be included).

Some of the experimental details are missing and need to be included prior to acceptance.

Finally, this reviewer noticed a couple of inappropriate citations that need to be corrected.

Additional minor revisions as detailed below.

Recommendation to editor: Accept with minor revisions.

Minor Points:

  1. Line 63: I would prefer the reference to the earlier work go after this sentence.

Answer: The reference has been cited as suggested by the reviewer (line 69).

  1. Line 66: Could the authors define “desirable techno-funcitonal properties” or list some of said properties in the manuscript?

Answer: As suggested by the reviewer, we have modified the text including examples of water and oil absorption capacities to emphasize this point (lines 67-68).

  1. Line 86: How was the freeze-dried sample reconstituted – in what buffer? How was the protein concentration determined – with what techniques / instrumentation? These are required details.

Answer: As suggested by the reviewer, we have provided more details about the reconstitution of the sample and the method used to quantify proteins (lines 89-95).

  1. Line 96: What is the composition of the “sample buffer” indicated here?

Answer: As indicated in the previous question, the description of the analysis has been extended including the information of the composition of the sample buffer (lines 89-95).

  1. Line 102: Ref [19] is inappropriate here. There are better citations describing this technique.

Answer: as suggested by the reviewer, we have substituted the reference [19] by a more suitable article describing the analysis (line 109).

  1. Line 111: Again after drying, what were the samples reconstituted in? and when desalted how was that done? In steps, with washes? What eluent? Volume? I assume they were redried again after desalting but that information is also missing.

Answer: As suggested by the reviewer, the description of the method has been extended (lines 118-125) as follows: “Then, they were resuspended in 50 µL of 0.1% TFA. For desalting, OMIX Pipette Tips C18 (Agilent Technologies, Santa Clara, CA, USA) were utilized, which were washed with 10 volumes of 100% ACN, followed by 10 volumes of 50% ACN. The tips were equilibrated in 0.1% TFA, and the sample was passed through the tips 10 times to ensure efficient binding. Subsequently, the tips were washed with 10 volumes of 0.1% TFA and eluted in 50 µL of a solution containing 0.1% TFA and 50% ACN, again passing the solution through the tips 10 times. After this process, the samples were dried once more …”.

Additionally, appropriate redaction and acronyms were established (lines 109, 110, 113, 114, 118, and 136).

  1. Line 113: How much of the sample was injected – representing how many ug of protein digest? How many injections total?

Answer: As suggested by the reviewer, the description has been extended for a better understanding (line 127-128). The dried, desalted protein digest was resuspended in 10 µL of 3% ACN and 0.1% formic acid, yielding approximately 2 µg of digested peptides derived from a total of 25 µg of protein.

Regarding the second point, only one injection was performed for this analysis. Despite this limitation, the findings adhere to the highest standards in proteomics. Each identification was rigorously filtered with a 1% false discovery rate (FDR), and proteins were confirmed by at least two unique peptides to ensure reliability. Additionally, de novo peptides met a strict score threshold (ACL > 85%). These results significantly enhance the Sacha Inchi proteome and underscore the robustness of our analytical approach.

  1. Line 125: Ref [21] is inappropriate here as it doesn’t fit to support the selection of orbitrap resolution.

Answer: Although the reviewer suggests deleting the references [21], we consider that this article is suitable for our analysis since it establishes the resolution of the analysis method for studies conducted in similar contexts.

  1. Line 229: Define the acronyms BP, CC, and MF in the text here.

Answer: The definitions of the acronyms BP, CC and MF can be found the first time they are cited (lines 168-169). Since they have been already defined, only the abbreviations can be read in line 247.

  1. Line 306: “to fulfill the nutritional needs human nutrition” change to “to fulfill human nutritional needs”

Answer: As suggested the sentence has been modified (line 325).

  1. Line 364: Sentence beginning here needs to be rewritten to make it clear the authors are describing previous results.

Answer:  As suggested by the reviewer, the text has been modified (line 381), changing “(data not shown)” by “(data found in the Peaks Project at the ProteomeXchange repository)”.

Additionally, we have rephrased the sentence (lines 384-386) as follows: “The specific expression of GAPDH in the species J. curcas, that belongs to the Euphorbiaceae family, resulted in a 3-4-fold increase in G3P, which ensured improved oil deposition...”.

  1. Line 396: delete “demonstrated”

Answer:  As suggested, the word “demonstrated” has been deleted.

Reviewer 3 Report

Comments and Suggestions for Authors

This article provides a new insight of the proteomic analysis of the major alkali-soluble Sacha Inchi (Plukenetia volubilis) proteins. The overall structure of the article is clear, with strong logical coherence, and the diagram can intuitively represent the content of the text description. However, there're a series of problems lacking innovation and normal form in this paper. This paper analysis frame is quite simple, possibly neglected some important analysis angle of view and the research content. In addition, there are still some issues in the article, please carefully confirm and answer the following questions.

1. This article exaggerates the result of the major alkali-soluble Sacha Inchi (Plukenetia volubilis) proteins by proteomic analysis, or rather describes it too broadly. It is recommended to describe it with clear priorities and emphasis.

2. It is difficult to to get useful and valuable information form the abstract. It is recommended to describe it with clear information, rather than a macroscopic point of view.

3. L20-21 “...., with numerous well-assigned spectra remaining unidentified.”, can the author give some explain to this?

4. This article transitions from the Gene Ontology (GO) annotations to the function of the major alkali-soluble Sacha Inchi (Plukenetia volubilis) proteins, but the connection between the two parts is not sufficient. What does the author purpose of GO analysis?

Author Response

Reviewer 3

This article provides a new insight of the proteomic analysis of the major alkali-soluble Sacha Inchi (Plukenetia volubilis) proteins. The overall structure of the article is clear, with strong logical coherence, and the diagram can intuitively represent the content of the text description. However, there're a series of problems lacking innovation and normal form in this paper. This paper analysis frame is quite simple, possibly neglected some important analysis angle of view and the research content. In addition, there are still some issues in the article, please carefully confirm and answer the following questions.

  1. This article exaggerates the result of the major alkali-soluble Sacha Inchi (Plukenetia volubilis) proteins by proteomic analysis, or rather describes it too broadly. It is recommended to describe it with clear priorities and emphasis.

Answer: We would like to thank the reviewer for his/her opinion. This study represents a rigorous proteomic analysis of Plukenetia volubilis proteins. To our knowledge, no similar reports describing proteins contained into this plant were found in the literature. Only, few proteins had been identified but any complete analysis of the proteome had been conducted. In the discussion, we highlighted the critical gaps existing in the current art state of Sacha Inchi protein characterization, connecting our findings to the broader context of the field. We believe this research makes a significant contribution to the current understanding of the Sacha Inchi proteome and offers valuable insights for its potential applications in the food industry. Some sentences have been added to the abstract, introduction and conclusions sections to remark the value of the obtained results from this study.

  1. It is difficult to get useful and valuable information form the abstract. It is recommended to describe it with clear information, rather than a macroscopic point of view.

Answer: As suggested by the reviewer, we have modified the abstract to provide clearer information about the findings of the study.

  1. L20-21 “...., with numerous well-assigned spectra remaining unidentified.”, can the author give some explain to this?

Answer: We would like to thank the reviewer by this point. We refer to the fact that the spectra corresponding to each identification were filtered with a stringent false discovery rate (FDR) of 1%. Furthermore, proteins identified by an unique peptide were meticulously reviewed manually to ensure reliability. Therefore, we have modified the text (line 381), substituting “(data not shown)” by ”(data found in the Peaks Project at the ProteomeXchange repository)”, which is now more appropriate. The readers accessing the Peaks Project can view these high-quality spectra filtered by an unique peptide.

  1. This article transitions from the Gene Ontology (GO) annotations to the function of the major alkali-soluble Sacha Inchi (Plukenetia volubilis) proteins, but the connection between the two parts is not sufficient. What does the author purpose of GO analysis?

Answer: GO analysis was conducted to elucidate the proteins' roles in seeds. The subsequent functional analysis was focused on identifying key proteins involved in storage, defense mechanisms, and lipid metabolism, underscoring their significance concerning potential applications in food science.

For a better understanding, we have revised the manuscript (abstract, and lines 274-275).
